# Camrelizumab combined with apatinib in patients with first-line platinum-resistant or PD-1 inhibitor resistant recurrent/metastatic nasopharyngeal carcinoma: a single-arm, phase 2 trial

Immunotherapy combined with antiangiogenic targeted therapy has improved the treatment of certain solid tumors, but effective regimens remain elusive for refractory recurrent/metastatic nasopharyngeal carcinoma (RM-NPC). We conducted a phase 2 trial to evaluate the safety and activity of camrelizumab plus apatinib in platinum-resistant (cohort 1, NCT04547088) and PD-1 inhibitor resistant NPC (cohort 2, NCT04548271). Here we report on the primary outcome of objective response rate (ORR) and secondary endpoints of safety, duration of response, disease control rate, progression-free survival, and overall survival. The primary endpoint of ORR was met for cohort 1 (65%, 95% CI, 49.6–80.4, $n = 40$) and cohort 2 (34.3%; 95% CI, 17.0–51.8, $n = 32$). Grade ≥ 3 treatment-related adverse events (TRAE) were reported in 47 (65.3%) of 72 patients. Results of our predefined exploratory investigation of predictive biomarkers show: B cell markers are the most differentially expressed genes in the tumors of responders versus non-responders in cohort 1 and that tertiary lymphoid structure is associated with higher ORR; Angiogenesis gene expression signatures are strongly associated with ORR in cohort 2. Camrelizumab plus apatinib combination effectiveness is associated with high expression of PD-L1, VEGF Receptor 2 and B-cell-related genes signatures. Camrelizumab plus apatinib shows promising efficacy with a measurable safety profile in RM-NPC patients.

Nasopharyngeal carcinoma (NPC) has unique geographical, etiological and biological characteristics that set it apart from other head and neck tumors[1]. It's mostly found in South China, Southeast Asia, the Middle East, and North Africa[1–3]. Nonkeratinizing NPC is the most frequent pathological subtype in endemic areas, the carcinogenesis of this subtype is closely related with Epstein-Barr virus (EBV) infection[4,5]. Patients with recurrent or metastatic NPC (RM-NPC) generally have a

poor prognosis, with a median overall survival (OS) of 20 months[6]. Platinum-based doublets, especially cisplatin plus gemcitabine(GP), is the standard first-line treatment for RM-NPC[7]. Immunotherapy, particularly with PD-1/PD-L1 inhibitor, has shown therapeutic efficacy in various cancers in recent year, as well as in RM-NPC. The 2022 NCCN Guidelines[8] and Chinese Society of Clinical Oncology (CSCO) clinical guidelines[9] had recommended that Cisplatin/gemcitabine(GP) plus

✉ e-mail: maihq@sysucc.org.cn

PD-1 inhibitor (pembrolizumab, nivolumab, camrelizumab, or toripalimab) be the first-line regimens for RM-NPC patients, respectively, based on the results of phase III clinical trial of CAPTAIN-1st and Jupiter-02 study[10,11]. Patients who are refractory or progress after chemotherapy plus PD-1 inhibitor treatment have few treatment options and currently there is no standard-of-care treatment[6]. In second-line therapy or later, the pembrolizumab/nivolumab and camrelizumab/toripalimab, with a limited objective response rate (ORR) ranging from 20.5% to 34.1%, have been approved by both the US Food and Drug Administration (FDA) and the Chinese National Medical Product Administration (NMPA)[12-15], respectively. However, subsequent randomized trials of anti-PD-1 monoclonal antibodies in second or later lines of treatment failed to demonstrate a survival benefit compared with chemotherapy alone[16,17]. Therefore, the development of exploratory synergistic combination therapies to improve the efficacy and overcome the resistance of PD-1 blockade for RM-NPC is urgently required.

The combination of a PD-1/PD-L1 inhibitor with an anti-angiogenesis antibody has shown efficacy in many malignancies[18-24]. Mechanistically, antiangiogenic agents can directly reduce regulatory T-cell proliferation and increase the infiltration of immune effector cells into tumors, enhance dendritic cell maturation and reprogram the tumor microenvironment by increasing vascular normalization[25-28]. Anti-angiogenics increased the density of high endothelial venules (typically surrounded by tertiary lymphoid structures) in the tumor microenvironment, which promoted T-cell trafficking to the tumor and overcame the endothelial immune cell barrier[27]. Second-line or later-line therapy with apatinib, an oral, small-molecule, tyrosine kinase inhibitor that selectively binds to VEGFR2[29], showed antitumor activity with acceptable toxicity and a response rate of ~30% in RM-NPC patients[30,31]. Phase II studies of the combination of camrelizumab and apatinib have been conducted in many solid tumors with encouraging efficacy and manageable safety[32,33]. However, currently the efficacy and safety of camrelizumab with apatinib in platinum-resistant and PD-1 inhibitor-resistant RM-NPC patients are still unknown.

In this phase 2 trial presented here, we report the results of camrelizumab plus apatinib as a second-line or later-line treatment regimen in platinum-resistant (cohort 1) and PD-1 inhibitor-resistant (cohort 2) RM-NPC patients. Both cohort 1 and 2 meets primary end point. In exploratory molecular analyses, B cell marker and tertiary lymphoid structure shows a positive associated with higher ORR in cohort 1, while a higher expression of PD-L1, VEGF Receptor 2 (KDR) and angiogenesis gene expression signatures are related with the higher efficiency in cohort 2. Overall, our study highlights the promising efficacy of the camrelizumab plus apatinib regimen as a viable treatment option for patients with platinum-resistant and PD-1 inhibitor-resistant RM-NPC. Moreover, our molecular analyses offer insights into potential biomarkers and mechanisms that could be targeted for further optimization of treatment strategies in this challenging disease setting.

## Results

### Patients

Between 8 September 2020, and 7 January 2021, 66 patients were screened, of whom 52 were enrolled in the study, with 27 in cohort 1 and 25 in cohort 2, all of whom were included in the full and safety analysis set. On Jan. 31, 2021, the protocol was amended to include 13 additional patients in the cohort 1 and 7 in the cohort 2, giving a total of 40 patients in cohort 1, 32 patients in cohort 2. One of the reasons for the protocol amendment was that too many patients had the desire to participate to this trial and that some of the patients consented before enrollment was halted because of encouraging results of previous enrolled patients in cohort 1 and cohort 2 (Supplementary Table 1 and Supplementary Fig. 1). The other reason was that principal investigator and statistician wanted to continue to evaluate the stability and

reliability of ORR and its 95% confidence intervals (CI) as the sample size expanded. All of amendment of the protocol was approved by the Research Ethics Board of Sun Yat-sen University Cancer Center (SYSUCC). The mean age was 45 years, and all patients had previously received at least first-line platinum-based treatment for RM NPC. There are 24 (33%) of the 72 patients from our previous published Jupiter 2 study[10], who suffered first-line treatment failure. All the patients in cohort 2 had been treated with the previous agent, a PD-1 inhibitor plus chemotherapy, or a bispecific antibody (BsAb) with dual targeting of PD-1 and CTLA-4 alone. In all, 51.4% of patients had liver metastasis, 19.4% were negative for PD-L1 (<1%), 52.8% were positive for PD-L1 (≥1%), and 20 (27.8%) patients had unknown PD-L1 status (according to PD-L1 TPS score). Baseline characteristics and previous treatment regimens are shown in Supplementary Table 1. There are 7 patients still on trial treatment at data off, and 65 are not receiving camrelizumab and apatinib treatment. Reasons for stopping treatment for the remaining 65 patients included disease progression (75.4%), toxicity (4.6%), completing 2 years of treatment (3.1%), or withdrawing from the study (16.9%). Eight patients in each cohort were excluded from the efficacy analysis due to the absence of a post-treatment efficacy assessment (Supplementary Fig. 1). There were three (4%) patients who dropped out of the study due to treatment-related adverse events, while eleven (16.9%) withdrew for other reasons. In cohort 1, a total of nine patients dropped out, with two owing to the COVID-19 pandemic. Of the remaining patients, four withdrew due to personal reasons, which included transportation difficulties and inability to afford examination costs. One patient died of pneumonia, and was not related to the study treatment. Two patients unexpectedly dropped out without citing any reason. Similarly, two patients in cohort 2 withdrew and refuse subsequent treatment due to personal reasons, choosing to resume local treatment because of transportation challenges. 36 (90%) of 40 patients in cohort 1 and 28 (87.5%) of 32 patients in cohort 2 were included in the efficacy analysis set.

### Efficacy

After enrollment of 27 and 25 patients in cohort 1 and cohort 2, respectively, considering that the primary study endpoint had been met for both cohorts, (cohort 1 with 18 responders and cohort 2 with 10 responders). By intention-to-treat analysis, of the 52 patients in the original cohort (i.e., before the protocol amendment), in cohort 1, objective responses were achieved in 18 (67%; 95% CI, 47.7–85.7) of 27 patients, and 22 patients had disease control (82% [65.8–97.1]) in the full analysis set. In cohort 2, objective responses were achieved in 40.0% (95% CI, 19.4–60.6) of 25 patients and 18 patients had disease control (72.0% [53.1–90.9]) in the full analysis set (Supplementary Table 2).

After the protocol amendment and inclusion of additional patients, the median time of follow-up at the time of data analysis (the data cutoff was 31 October 2022) was 23.3 months (IQR 22.4–24.2 months) for cohort 1 and 18.5 months (17.3–19.7 months) for cohort 2. In cohort 1, 3 (7.5%) of 40 patients had a confirmed complete response, 23 (57.5%) had a confirmed partial response, 6 (15.0%) had stable disease, and 4 (10.0%) had progressing disease in the full analysis set. In total, objective responses were achieved in 26 (65%; 95% CI, 49.6–80.4) of 40 patients, and 32 patients had disease control (80% [67.0–93.0]) in the full analysis set (Fig. 1A and Supplementary Table 3). In the efficacy analysis set, 26 (72.2% [95% CI 56.9–87.6]) of 36 patients had an objective response, and 32 (88.9% [78.1–99.7]) had disease control. We also evaluate the stability of ORR and its 95% confidence intervals (CI) in cohort 1 as the sample size increased, and we found that ORR and its 95% CI presented good stability followed the patients number expanded (Supplementary Fig. 2A). The median duration of response was 14.6 months (95% CI 5.5–NE), and post-hoc analysis showed that the 12-month duration of response rate was 53.8% (95% CI 34.6–73.0) (Fig. 1E and Supplementary Table 3). Tumor shrinkage was noted in 32 (89%) of 36 patients who had

at least one efficacy assessment. The mean best percentage change of the target lesion size from the baseline was −40.7% (SD 0.27) (Fig. 1B). In the prespecified exploratory analysis, among 40 patients, 23 (57.5%) were PD-L1+ and 6 (15.0%) were PD-L1- defined by SP142

Immunohistochemistry (IHC) staining, whereas eleven (27.5%) patients had unknown PD-L1 status. PD-L1+ patients, defined by TPS positive staining>1%, had numerically higher ORR than PD-L1- patients (65.2% vs 50.0%), but the difference was not statistically significant

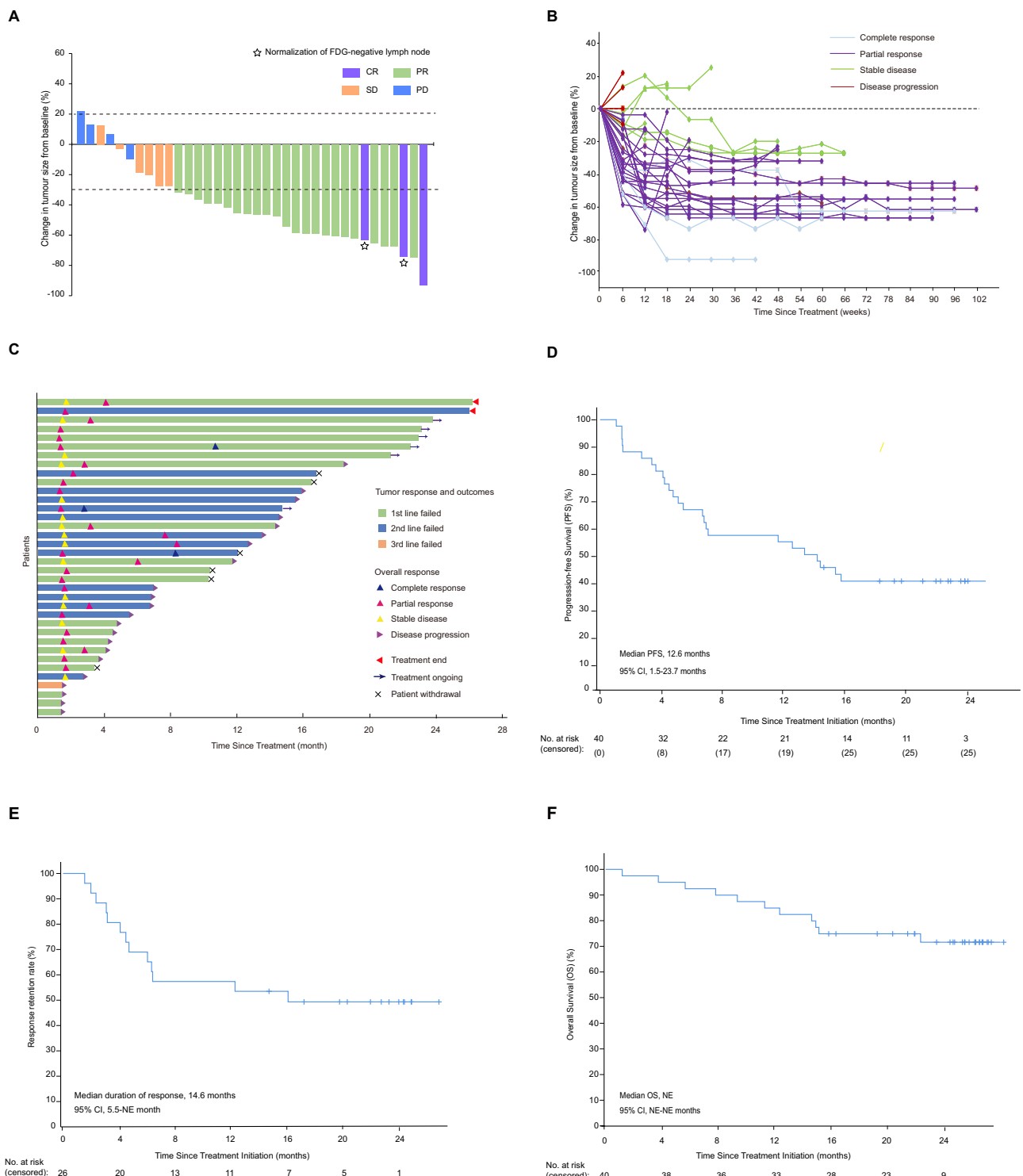

**Fig. 1 | Overall response and survival of platinum-resistant NPC patients (cohort 1). A** Waterfall plot of the best response in platinum-resistant NPC patients. Best change in the sum of target lesion size compared with that at baseline (*n* = 36). The horizontal lineat −30 shows the threshold for defining an objective response in the absence of non-target disease progression or new lesions according to RECIST 11. One patient with 100% reduction in target lesion size had non-target lesions present. White stars, two patients had normalization

(<10 mm) of fluor-8-deoxyglucose (FDG)-negative lymph nodes (at baseline, lymph nodes were >1.5 cm and FDG+) and 100% reduction of non-lymph node lesions and are considered to have had a CR (purple bar). **B** Spider plot of measurements of target lesions at each timepoint in platinum-resistant NPC patients (*n* = 36). **C** Swimmer plot (*n* = 36). **D** Kaplan–Meier curves for progression-free survival. **E** Response retention rate in platinum-resistant NPC patients. **F** Kaplan–Meier curves for overall survival.

(*P* = 0.65) (Supplementary Fig. 4A, B). To comprehensively evaluate the correlation between PD-L1 expression and ORR, we simultaneously conducted CPS scoring. Results show that 65.0% were PD-L1+, 7.5% were PD-L1-, and 27.5% had an unknown status. Patients positive for PD-L1 (defined by CPS positive staining>1) had numerically higher overall response rates (ORR) than those who were PD-L1 negative (65.4% vs 33.3%), although the difference was not statistically significant (*P* = 0.53; Supplementary Fig. 4C). As determined by VEGF receptor 2 (KDR) IHC staining in cohort 1, 21 (52.5%) of the 40 patients in the prespecified exploratory analysis, were KDR+ and 8 (20.0%) were KDR-, whereas 11 (27.5%) patients had unclear KDR status. The ORR was numerically larger in KDR+ patients than in KDR- patients (71.4% vs. 37.5%), who were identified by positive staining of >1%, but the difference was not statistically significant (*P* = 0.20; Supplementary Fig. 5A, B).

In cohort 2, 11 (34.4%) of 32 patients had a confirmed partial response, 11 (34.4%) had stable disease and 6 (18.8%) had progress disease. In total, objective responses were achieved in 34.4% (95% CI, 17.0–51.8) of 32 patients and 20 patients had disease control (68.8% [51.8–85.7]) in the full analysis set (Fig. 2A and Supplementary Table 3). In the efficacy analysis set, 11(39.3% [95% CI 20.0–58.6]) of 28 patients had an objective response, and 22(78.6% [62.4–94.8]) had disease control. The stability of ORR and its 95% CI in cohort 2 was analyzed and good stability was also observed as the number of enroll patients increased (Supplementary Fig. 2B). The median duration of response was 2.9 months (95% CI 1.4–4.5), and post-hoc analysis showed that the 6-month duration of response rate was 36.4% (95% CI 8.0–64.8; Fig. 2E). All of the 11 patients who achieved an objective response had progressed or died before the data cutoff. Tumor shrinkage was observed in 16 (57%) of the 28 patients who had at least one post-baseline efficacy evaluation. The best percentage change in the size of the target lesion from baseline was −12.0% (SD 0.31). (Fig. 2B). Regarding post-hoc analysis of 32 patients, 15 (46.9%) had PD-L1 positivity, 8 (25.0%) had PD-L1 negativity, and 9 (28.1%) had uncertain PD-L1 status. PD-L1+ patients had greater ORR than PD-L1- patients, although the difference was not statistically significant (*P* = 0.18; Supplementary Fig. 4B; according to PD-L1 TPS score). For the PD-L1's CPS, in the post-hoc analysis of 32 patients, 56.3% were PD-L1+, 15.6% were PD-L1-, and 28.1% had an unknown PD-L1 status. Patients positive for PD-L1 had a greater ORR than PD-L1- patients, but the difference was not statistically significant (*P* = 0.62) (Supplementary Fig. 4C). Fifteen (46.9%) had positive KDR results, as assessed by positive staining >1%, compared to 8 (25.0%) who had negative results, and 9 (28.1%) who had ambiguous KDR results. KDR+ patients had a higher ORR than KDR- patients (46.7% vs. 12.5%), although the difference was not statistically significant (*P* = 0.18; Supplementary Fig. 5B). To monitoring other targets of apatinib, the expression of c-KIT and SRC were detected in each arm. Responders in cohort 2 were characterized by higher c-KIT expression; however, no difference was observed in responders versus non-responders in cohort 1. The expression of SRC did not differ between response and non-response group in both cohorts, because its high-level expression in situ and metastatic tissues of nasopharyngeal carcinoma, the percentage of SRC positive cell was over 75% in most tissues (Supplementary Fig. 6A, B).

## Survival

In cohort 1, 34 (85.0%) of 40 patients had discontinued the treatment by the cutoff date, and 6 (15.0%) patients remained on treatment (Fig. 1C and Supplementary Fig. 1). Twenty-two (64.7%) of the 34 patients were discontinued because of disease progression, and one (2.9%) patient was discontinued as a result of adverse events. Other reasons for treatment discontinuation included completing 2 years of treatment (two [5.9%]) and withdrawal of consent (nine [26.5%]). 25 progression-free survival events occurred (14 patients had disease progression and 11 patients died), the median progression-free survival was 12.6 months (95% CI 1.5–23.7) and the median overall survival was

not reached. 1-year overall survival was 82.5% (95% CI 70.7–94.3; Fig. 1D, 1F, and Supplementary Table 3). At the last data cutoff for follow-up, 28 patients were alive and six remained on the study. 11 deaths occurred: 10 due to disease progression and 1 due to pneumonia.

In cohort 2, 31 (97%) of the 32 patients had dropped out, and 1 (3%) was still on treatment (Fig. 2C and Supplementary Fig 1). Twenty-seven (87%) of the 31 patients dropped out of the research due to disease progression, while 2 (7%) withdrew due to adverse events, 2 (7%) withdrew from the study. 28 progression-free survival events occurred (15 patients had disease progression and 16 died), and the median progression-free survival was 4.5 months (95% CI: 3.7–5.4, Fig. 2D), and the median overall survival was 16.2 months (95% CI 13.1-NE; Fig. 2F). 1-year overall survival was 68.8% (95% CI 58.9–91.1). At last follow-up, 16 patients were alive and one remained on study. 16 deaths were due to disease progression.

Exploratory subgroup analysis of the full analysis set for cohorts 1 and 2 showed that the patients with liver metastasis had a similar ORR compared with the patients without. Patients who had previously been treated with EGFR inhibitors had a higher ORR than those who had not been treated with EGFR inhibitors in both cohort 1 (72.7% vs 62.1%) and cohort 2 (57.1% vs 24.0%). A higher ORR was also observed in patients with EBV DNA titers <10,000 copies/ml than in patients with >10,000 copies/ml in both cohort (73.1% vs 50.0%)1 and cohort2(42.9% vs 18.2%). But the difference was not statistically significant (*P* = 0.18, *P* = 0.14; Supplementary Tables 6). Dynamic monitoring of plasma EBV DNA copy number was performed during the study, and results were available for 72 patients. In both cohorts, patients with objective responses had greater decreases in EBV titers from baseline to day 28 compared to patients with stable or progressive disease (Supplementary Fig. 7). Furthermore, patients with ≥50% EBV titer decrease on day 28 had a significantly higher ORR than those with <50% decrease in both cohort 1 (70% vs 60%) and cohort 2 (43.8% vs 25.0%). All of subgroup analysis of efficacy were detail presented in Supplementary Tables 6 and 7.

## Safety

In the safety set, dose reductions occurred in 49 (68.1%) of 72 patients for apatinib, of whom 39 (79.6%) patients required only one level of dose reduction and 17 (34.7%) patients had two levels of dose reduction. Treatment-related adverse events led to dose interruptions of apatinib in 49 (68.1%) of 72 patients, and of camrelizumab in 15 (20.8%) patients. The most common reasons for interruptions of apatinib were hypertension (19 [38.8%]), hand and foot syndrome (15 [30.6%]), increased aspartate aminotransferase (nine [18.4%]), and increased alanine aminotransferase (eight [16.3%]); similarly, increased aspartate aminotransferase (nine [18.4%]) and increased alanine aminotransferase (eight [16.3%]) led to camrelizumab interruptions. Nine patients discontinued apatinib because of nasopharyngeal necrosis (seven [77.8%]), increased alkaline phosphatase (one [11.1%]), and thrombocytopenia (one [11.1%]). Three patients discontinued camrelizumab because of rash (two [66.7%]) and immune myocarditis (one [33.3%]).

The most common grade 3–4 adverse events were hypertension (27.8%), hand-foot syndrome (12.5%), and the increase in AST (11.1%). Serious adverse events were reported in three patients, with two presenting with rash, and one with acute immune myocarditis, all of which were considered treatment-related. In nine (12.5%) of patients, receptive cutaneous slender endothelial multiplication (RCCEP), a common and self-limiting trAE of camrelizumab, was observed, with two individuals revealing grade 3 events. RCCEP occurred only on the skin, with pathology revealing hairlike endothelial hyperplasia and slender hyperplasia in the dermis. During the treatment period, we saw the occurrence of nasopharyngeal necrosis in 9 cases (12.5%) and the diagnosis for all cases of nasopharyngeal necrosis was confirmed

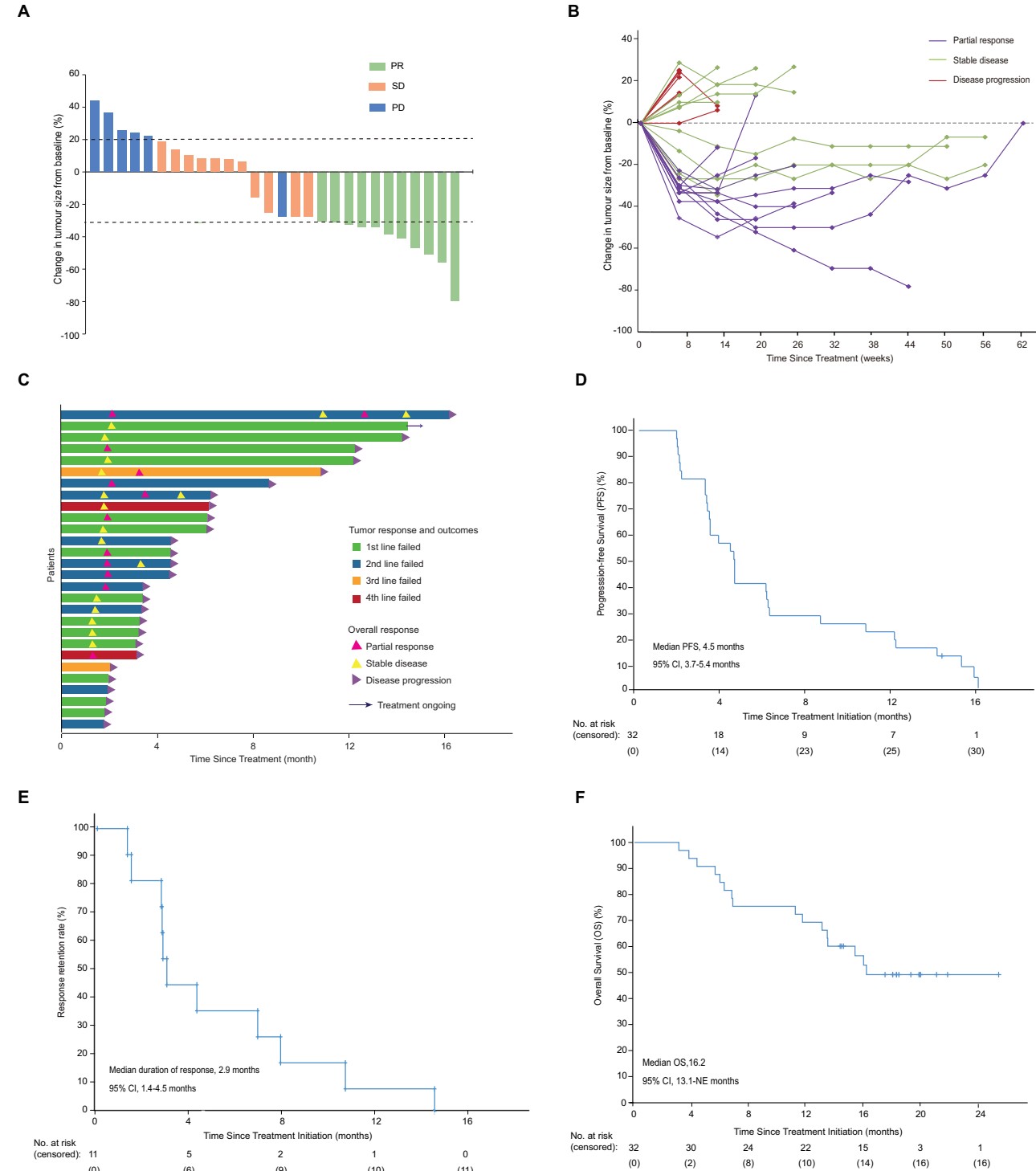

**Fig. 2 | Overall response and survival of PD-1 antagonist-resistant NPC (Cohort 2). A** Waterfall plot of the best response in PD-1 antagonist-resistant NPC patients (*n* = 28). **B** Spider plot of measurements of target lesions at each time point in PD-1 antagonist-resistant NPC patients (*n* = 28). **C** Swimmer plot (*n* = 28). **D** Kaplan–Meier curves for progression-free survival. **E** Response retention rate in PD-1 antagonist-resistant NPC patients. **F** Kaplan–Meier curves for overall survival.

through nasopharyngoscope and/or magnetic resonance imaging. All nine patients with nasopharyngeal necrosis were classified as grade 3 or higher (Supplementary Table 4), and one patient with grade 4 experienced massive hemorrhage. All nine patients with nasopharyngeal necrosis received hyperbaric oxygen and debridement therapy and no one dead of hemorrhage. The detailed information regarding these nine patients was shown in Supplementary Table 8. In cohort 2, with apatinib alone throughout the neoadjuvant phase, we also

counted the cumulative acute adverse events, and the most frequent adverse events were increased AST (15 [46.9%]), reduced leukocytes (12 [37.5%]), and increased ALT (11 [34.4%]) (Supplementary Table 5).

## B cells and tertiary lymphatic structure found in the tumors of responders in cohort 1

To gain insight into the mechanisms of therapeutic responses as well as biomarkers of response and resistance, longitudinal tumor samples,

including baseline (before combination therapy) and relapse, were taken in the context of therapy, and molecular and immune profiling was performed (Supplementary Fig. 8). We performed RNA sequencing (RNA-seq) in longitudinal tumor samples within each treatment cohort and across treatment cohorts. According to the analysis results of differentially expressed genes, significantly higher expression of B-cell-related genes such as *MZB1, JCHAIN,* and *IGHL* was observed in patients that respond to camrelizumab plus apatinib treatment versus non-responding patients ('responders' $n = 9$ and 'non-responders' $n = 4$, hereafter) at baseline ($P < 0.01$) with over-representation of these genes compared to T cells and other immune markers (with evaluable tumors from seven responders and nine non-responders) in cohort 1 (Fig. 3A, B and Supplementary Table 9). KEGG enrichment analysis of differentially expressed genes showed that B-cell-mediated immunity was important for treatment response (Supplementary Fig. 9A). Other genes that are expected to alter the function of B cells were also significantly enriched in responders versus non-responders, such as *BTLA* (Supplementary Data 1). Low tumor purity was observed in some samples, particularly in the context of an effective therapeutic response, limiting conventional analysis of RNA-seq data. To address this, we next performed a more focused investigation of the tumor immune microenvironment using the microenvironment cell populations (MCP)-counter method on RNA-seq data in baseline and relapse tumor samples-focusing more specifically on immune-related genes, which allowed inclusion of samples with low tumor purity (12 responders and 5 non-responders at baseline) (Supplementary Table 9 and Supplementary Data 3). We again observed enrichment of a B cell signature in responders versus non-responders at baseline ($p = 0.004$). Notably, this analysis included samples from patients with nodal and extra-nodal lesions with no obvious contribution based on the site of disease, which suggests that B cell signatures were not merely related to the presence of these tumors within lymph nodes (Fig. 3C and Supplementary Data 2). High objective response rate (ORR) was observed in high B cells, T cells, cytotoxic lymphocytes, and myeloid dendritic cell populations (Fig. 3D). B cell signatures alone were predictive of response in univariable analyses (odds ratio 1.38, $P = 0.035$) for our trial, but not in multivariable analyses when considering other components of the immune cell infiltrate, which suggests that B cells probably act together with other immune subsets and are not acting in isolation. However, these analyses were limited owing to the low sample size (Fig. 3E and Supplementary Table 10). In order to confirm the correlation between B lineage and clinical prognosis of nasopharyngeal carcinoma patients, we applied the MCP-counter algorithm to available RNA-seq data from the additional locoregionally advanced 98 NPC patients (Supplementary Table 12 and 13) and compared overall survival in patients with tumors high for B cell lineage versus low, which demonstrated prolonged overall survival in patients with B cell-lineage-high tumors ($P = 0.047$; Supplementary Fig. 10A, B).

The relevance of MCP-counter scores for B lineage and these cell types showed that there was a high correlation between B lineage and T cells in cohort 1 (Fig. 3F). In contrast to other immune cells present in the TME, intratumoral B cells are mostly associated with the tertiary lymphatic structure (TLS). On the basis of the results from gene expression profiling, we next assessed tumor samples histologically to gain insight into the density and distribution of B cells as well as their relationship to TLSs in patients treated with camrelizumab plus apatinib. Illustration of the spatial organization of T cells and B cells in the tumor microenvironment (TME) based on immunohistochemical staining of primary and metastatic lung or liver lesion for CD19 (B cell marker; brown) and CD3 (T cell marker; rose red), TLSs were confirmed to exist in primary and distant metastatic lesion (Fig. 3H). Multiplex immunofluorescence assay of TLSs showed that the density of CD19 + B cells and TLS mean area were higher in responders than in non-responders (Fig. 3G, I). B cells can be efficient antigen-presenting

cells (APCs), particularly when they are in close proximity to memory CD4 + T cells, as is the case in TLS. Proximity analysis between CD19 + B cell and CD4 + T cell was performed with Halo analysis software (PANOVU). B cells are closer to CD4 + T cells in the TLS than in the dispersed area (Supplementary Fig. 11A–C). Together, these data provide insights into the potential role of B cells and tertiary lymphoid structures in the response to immune checkpoint inhibitors combined with antiangiogenic targeted therapy in RM NPC, with implications for the development of biomarkers and therapeutic targets.

**Angiogenesis and blood vessel density are predictive of clinical response to camrelizumab plus apatinib in cohort 2**
Given that the response rate of camrelizumab plus apatinib in cohort 2 (PD-1 inhibitor-resistant) was much lower compared with cohort 1 (PD-1 inhibitor-naive), suggesting that there is significant heterogeneity among patients in these cohorts, possibly due to variances in immune microenvironments and vascular density. These factors contribute to the variability in patient response observed in our study. In order to explore the mechanism of PD-1 inhibitor resistance in patients with advanced nasopharyngeal carcinoma, we analyzed the baseline sequencing data in two cohorts. Compared with the patient in cohort 1, differentially expressed genes such as *MYL2, MYLPH* were mainly enriched in muscle contraction related pathways (Supplementary Fig. 12A–C and Supplementary Data 4). MCP analysis suggested neutrophils was significantly increased in cohort 2 (Supplementary Fig. 12D and Supplementary Data 3). Subsequently, we explored predictive biomarkers for the patients treated with camrelizumab plus apatinib in cohort 2. A heatmap of genes previously defined and representing angiogenesis and immune biology in 13 evaluable pre-treatment tumors (eight responders and five non-responders) showed different signature scores based on relative expression levels of angiogenesis (Angio), immune and antigen presentation (Immune), and myeloid inflammation-associated genes (Myeloid)[34–38] (Fig. 4A and Supplementary Table 15). High Angio signature score, but not immune and myeloid signature, was found in responders (Fig. 4B and Supplementary Table 11). High expression of the Angio gene signature, based on median signature score, was associated with improved ORR (87.5% in Angio^High versus 20% in Angio^Low) (Fig. 4C, D). Responders in cohort 2 were characterized by higher vascular density as evaluated by CD31 IHC; however, no difference was observed in responders versus non-responders in cohort 1 (Fig. 4E, F). In addition, angiogenesis markers VEGFA and IL-8 were also detected in each arm, and higher expression was validated in the response group of both cohorts (Supplementary Fig. 13A, B). This suggests that angiogenesis markers can be used as biomarkers to predict the efficacy of combination therapy. Several studies have shown that vascular endothelial cells may express PD-L1 in specific situations and that its expression can be induced by immune stimulation[39–41]. We conducted CD31-labeled vascular endothelial cell (red) and PD-L1 (green) immunofluorescence staining on tissue paraffin sections. The percentage of co-expressed cells was higher in responders for both cohorts. However, the overall low co-expression and limited sample size could potentially compromise statistical significance (Supplementary Fig. 14A–C). MCP analysis showed that the immune microenvironment was similar in responders and non-responders (Fig. 4G, H).

To separate the benefit of apatinib and its therapeutic pathways in target, we detected transcriptomic CD8 or IHC CD8 and angiogenesis markers in one arm versus another. We observed no variation in transcriptome CD8A and CD8B between the two cohorts; however, IHC CD8 staining showed higher values in cohort 1 (Supplementary Fig. 15A). This finding may explain the cause of PD1 inhibitor resistance in cohort 2, since the immune microenvironment is in a suppressive state. No significant difference in CD8 expression existed between response and non-response, indicating that CD8 + T cells do not play a major role in determining treatment efficacy

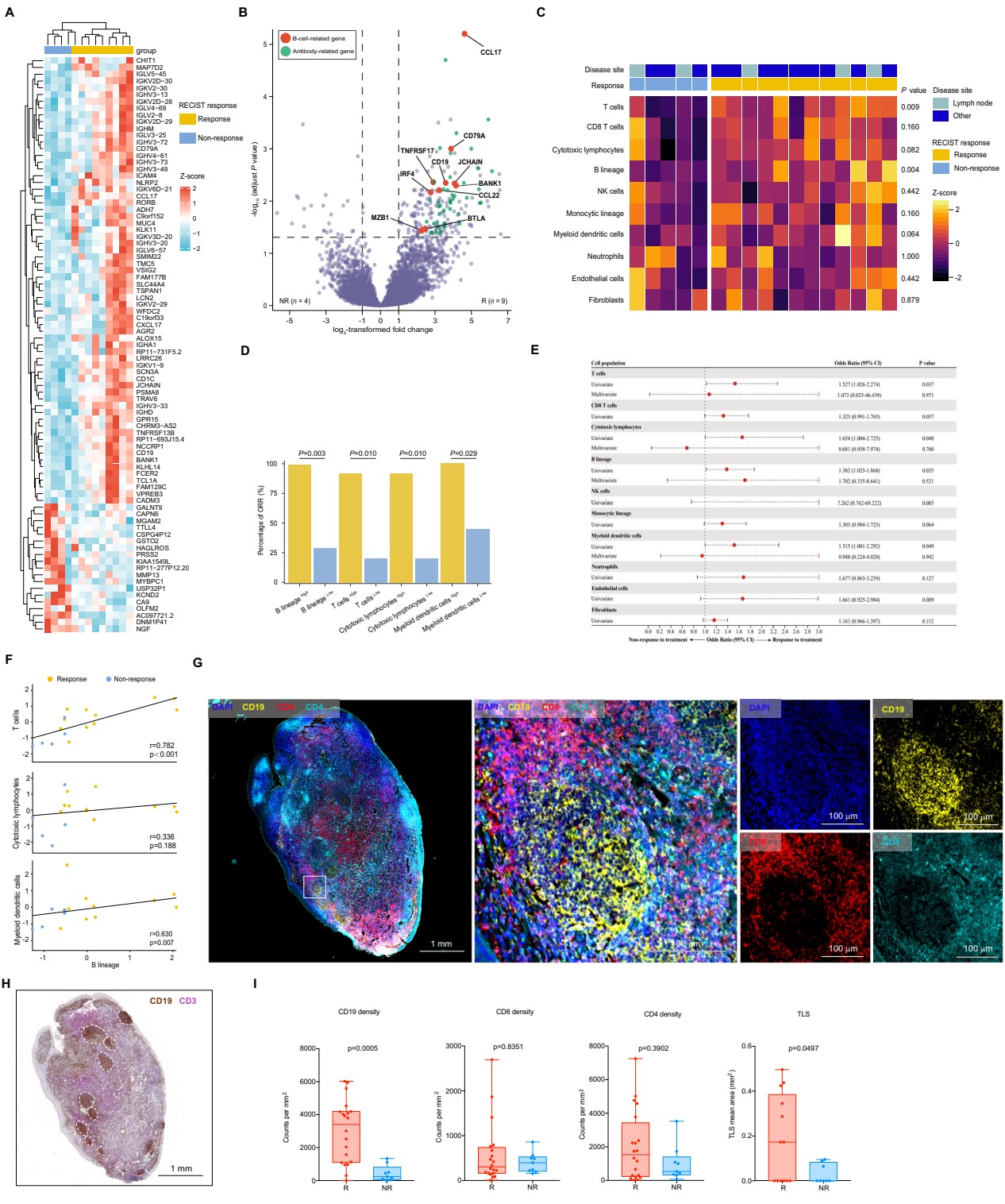

(Supplementary Fig. 15B). Our study revealed that B lymphocytes and tertiary lymphoid structures are essential for achieving treatment response in platinum-resistant patients. For cohort 2, individuals with high vascular density are more likely to experience positive outcomes from treatment, underscoring the importance of anti-angiogenic therapy. We tested angiogenesis markers in both cohorts and found no differences in CD31, VEGFA, and IL8 levels between them (Supplementary Fig. 15C). However, expression of these markers was elevated in responders, suggesting the efficacy of apatinib (Supplementary Fig. 15D).

## Evolution of tumor immune contexture at relapse

We interrogated the TME dynamic changes of camrelizumab plus apatinib in treated patients developing secondary treatment resistance. 11 paired samples were collected at baseline and after relapse (responder $n = 7$, non-responder $n = 4$). We found a significant decrease in KDR in patients with tissue matching before and after treatment ($p = 0.004$; Supplementary Fig. 5C). For delayed resistance (responders, $n = 7$) to anti-PD-1 and anti-VEGF combination therapy, the TME at relapse evolved toward an increase in expression of oxidative phosphorylation and ATP synthesis was observed at relapse,

**Fig. 3 | TLSs containing B cells are predictive of clinical response to camrelizumab plus apatinib in platinum-resistant NPC patients. A** Supervised hierarchical clustering of differentially expressed genes (DEG) on RNA-seq analysis by response of nasopharyngeal carcinoma tumor specimens at baseline, with responder defined as having a complete or partial response by RECIST 1.1 and nonresponder as having less than partial response (*n* = 4 non- responders and 9 responders). A cut-off of gene expression fold change of ≥2 or ≤−2 and a false discovery rate (FDR) *q* ≤ 0.05 was applied to select DEGs. **B** Volcano plot of differentially expressed genes from baseline tumor specimens by response (*n* = 4 nonresponders and 9 responders). The *x*-axis in volcano plots depicts log2-transformed fold changes (FC) of Rs versus NRs. The vertical dashed lines represent the log2-transformed FC thresholds of 1 or −1. The horizontal dashed line represents the two-sided DESeq2-adjusted *P* value threshold of 0.05. B-cell-related genes (red) and antibody-related genes (green) are distinguished by colors. Both genes met the following criteria: log2-transformed FC> 2 or <−2 and a DESeq2-adjusted *P* value of <0.05. **C** Association between objective response rate (ORR, percentage) and MCP score at cutoff mean values, *P* values calculated by two-sided Fisher's exact test. **D** Unsupervised hierarchical clustering analysis shown for baseline tumor specimens by response (*n* = 12 Rs and 5 NRs). Unique clusters are indicated by green color on top row. *P* values were calculated by two-sided Mann–Whitney *U* test. **E** Forest plots of response odds ratios (ORs) and confidence interval (CIs) for MCPcounter. All OR and CI values for response were extracted from logistic regression models. **F** Relevance of MCP-counter scores for T cells, cytotoxic lymphocytes and myeloid dendritic cells with regard to B lineage (*n* = 12 Rs and 5 NRs) as indicated. Correlation coefficients were calculated by Spearman correlation analyses. Each dot represented one sample. **G** Multiplex immunofluorescence assay of TLSs for the following markers: CD19, CD4, CD8, and DAPI. Original magnification, ×20. Scale bar is 1 mm or 100 μm. **H** Representative image of CD19 and CD3 staining of TLSs in a responder after treatment with camrelizumab combined with apatinib. Scale bar is 1 mm. **I**. Quantification of CD19, CD4, CD8 (cohort 1, *n* = 20 Rs and 9 NRs) and TLSs (cohort 1 nonlymph node tissues, *n* = 13 Rs and 8 NRs) density by multiplex immunofluorescence assay and association with response to camrelizumab combined with apatinib in patients with first-line platinum-resistant. All data in box and whiskers plots (3I) are represented as median value, quartile, maximum and minimum; each dot represented one sample. *P* values were calculated by two-tailed, Mann–Whitney *U* test. Source data are provided as a Source Data file.

including genes *COX4I1, COX6A1, DLD, and NDUFA2* (Supplementary Fig. 16A–C and Supplementary Data 5). Concurrently, an increase in extracellular matrix organization compared to baseline TME was observed at relapse, including genes involved in collagen fibril organization, *AEBP1, COL1A1*, and *COL11A1* (Supplementary Fig. 16D and Supplementary Data 5). Consistent with these findings, the MCP analysis result showed a significant expansion of fibroblasts at relapse (Fig. 5A, B and Supplementary Data 3). Fibroblasts are the main source of collagen, which is important for the formation of extracellular matrix. The multiplex immunofluorescence of paraffin sections was consistent with the sequencing results, and the number of fibroblasts in the relapse tissues increased significantly (Fig. 5C, D). Besides, the microenvironment cell populations analyze suggested that drastically decreased B lineage was observed at relapse, suggests that B lineage played an important role in therapeutic response of camrelizumab combined with apatinib.

For intrinsic resistance (non-responders, *n* = 4) to anti-PD-1 and anti-VEGF combination therapy, signal pathways related to muscle system process are enriched in relapse tissues compared with baseline, the expression of *ACTA1, MYH7, TNNC2,* and *TNNT3* were increased (Supplementary Fig. 16E–G and Supplementary Data 6). MCP analysis showed that fibroblasts increased at relapse for non-responders (Fig. 5A, B and Supplementary Data 3). Together, these data suggest that the treatment process may stimulate the expansion and proliferation of fibroblasts, leading to the increase of collagen secretion and strengthening of fibrosis, forming a physical barrier in the process of drug resistance and preventing the drug from effectively reaching the target.

## Discussion

The combination of immune checkpoint inhibitors with antiangiogenesis drugs has previously shown synergistic efficacy in patients with several types of solid tumors[21–24,32,33,42], and our trial aims to assess the activity and safety of combining an anti-PD-1 antibody with an angiogenesis inhibitor as second or later-line treatment for RM-NPC patients. Respective of PD-L1 expression status, camrelizumab in combination with apatinib exhibited encouraging antitumor activity in platinum-resistant (cohort 1) and in PD-1 inhibitor-resistant (cohort 2) RM-NPC patients. Camrelizumab plus apatinib had objective response rates of 65.0% and 34.4% and a median PFS of 12.6 months and 4.5 months in cohort 1 and 2, respectively, with manageable treatment-related adverse events. In platinum-resistant disease, the proportion of patients achieving an objective response with camrelizumab and apatinib was greater than that reported with single-agent apatinib or PD-1 inhibitors, with an ORR from 17.0% to 34.1%[6,10–15,29–31]. Compared to these studies, our treatment effectiveness (ORR at 65.0%) showed a

significant improvement (*P* < 0.01; Supplementary Fig. 17) and PFS showed a notable improvement of 7 to 10 months. Furthermore, the 1-year overall survival rate in this cohort was significantly higher at 82.5% (95% CI 70.7–94.3) than that achieved with previous monotherapy (34.3–63%) (Supplementary Table 14). In PD-1 inhibitor-resistant disease, the proportion of patients achieving an objective response with camrelizumab and apatinib was still higher than that reported with apatinib monotherapy, with an ORR round to 30%[29–31]. These findings indicated that camrelizumab in combination with apatinib could be a potential therapeutic option for the RM-NPC patients who suffered first-line platinum-or PD-1 inhibitor-based treatment regimen.

Despite of a high proportion of patients with RM-NPC can achieve a response with GP plus toripalimab or camrelizumab regimens during first-line treatment[10,11], according to our previous published phase III trial. Salvage chemotherapy, targeted drugs, and PD-1 inhibitor monotherapy only produce moderate antitumor activity as second-line or later treatments for this patient population[14]. The randomized phase 3 trials proved that second or later-line anti-PD-1/PD-L1 monotherapy does not prolong median progression-free survival compared with chemotherapy in RM-NPC patients from Keynote 122 and Even's study[16,17,43]. According to previous published trials[10–15], median progression-free survival is 1.9–5.6 months in patients receiving anti-PD-1 monotherapy as the second or later-line treatment. In this study, combination treatment with camrelizumab and apatinib resulted in a median progression-free survival of 12.6 months for platinum-resistant RM-NPC patients. Even for the patients with PD-1 inhibitor resistant, the PFS with combination of camrelizumab and apatinib was still 4.5 months, which was never reported in previous study, in spite of the PFS of apatinib monotherapy was 3.9 months for PD-1 inhibitor naïve patients. PD-L1 and VEGF receptor 2 (KDR) expression is the most extensively used biomarkers with respect to predicting the efficacy of immune checkpoint inhibitors and angiogenesis inhibitors, respectively. Consistent with nivolumab and toripalimab in RM-NPC, our study observed a numerically higher but not statistically significant ORR of PD-L1 positive patients than PD-L1 negative patients between TPS and CPS, as well as for KDR positive than KDR negative patients both in cohort 1 and cohort 2 patents. In addition, the combination of EGFR inhibitors and immunotherapy has demonstrated promising therapeutic effects on head and neck tumors[44,45]. Our subgroup analysis revealed a higher ORR in patients receiving EGFR inhibitors. These findings are in line with previous research.

This study reported a similar safety profile of camrelizumab plus apatinib treatment that was consistent with previous studies for other solid tumors. The most common treatmentrelated adverse events of any grade were leukopenia, neutropenia, fatigue, proteinuria, hand

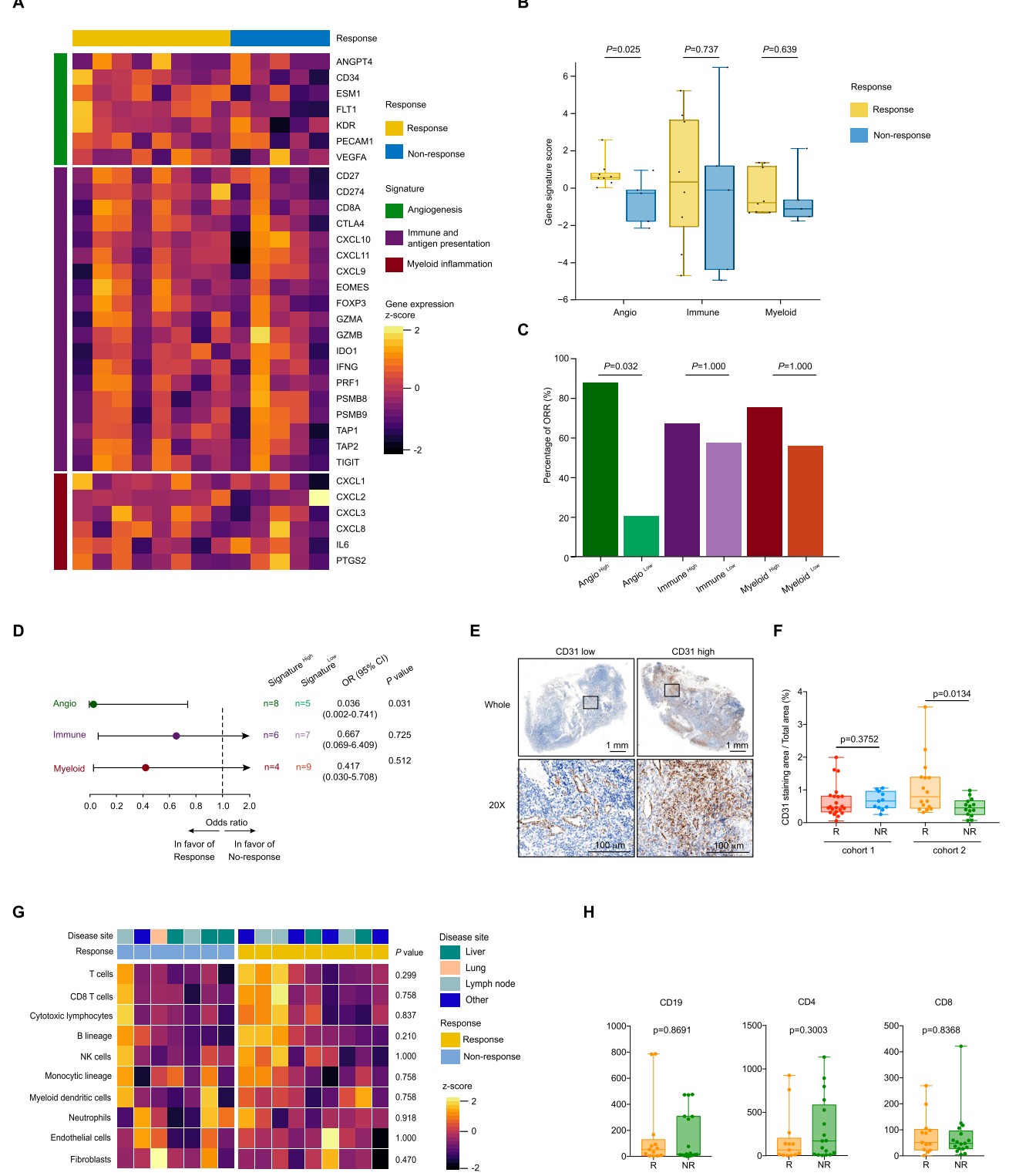

and foot syndrome, increased ALT and AST level, and hypothyroidism. Notably, compared with camrelizumab monotherapy[14], the incidence of reactive cutaneous capillary endothelial proliferation was greatly decreased. The most common grade 3 treatment-related adverse event was hypertension, hand and foot syndrome, liver transaminase level increased and nasopharyngeal necrosis, which could be resolved by an interruption or reduction of apatinib dose. Most of the naso-pharyngeal wall necrosis recovered to normal after the interruption of apatinib and was treated with conservative debridement and hyper-baric oxygen therapy. Of the 9 patients with nasopharyngeal necrosis,

only 1 patient had a local recurrence of the disease after combination therapy, while the rest were diagnosed with metastatic nasophar-yngeal cancer. Considering the founding of our study and previous report[46–48], the combination therapy of camrelizumab and apatinib will increases the risk of nasopharyngeal necrosis, but the toxicity was manageable and the nasoendoscopy should be performed every two cycles. In our research, 45 individuals (62.5%) previously underwent radiotherapy, and 7 of them developed necrosis. Those who experi-enced necrosis had a shorter time between the completion of radio-therapy and beginning anti-angiogenic therapy compared to those

**Fig. 4 | Angiogenesis and blood vessel density are predictive of clinical response to camrelizumab plus apatinib in PD-1 antagonist-resistant NPC patients. A** Heatmap showing the gene expression profiles of signatures (rows) in baseline tumor specimens grouped by response ($n$ = 8 Rs and 5 NRs). Normalized gene expression data related to angiogenesis (green), immune and antigen presentation (purple), and myeloid inflammation (red) were converted using $z$-score method for visualization. **B** Associations between response ($n$ = 8 Rs and 5 NRs) and signature scores. Gene signatures were defined as follows: angiogenesis (Angio); immune and antigen presentation (Immune); myeloid inflammation (Myeloid). Box plots show $z$-normalized scores for response categories, with $P$ values calculated by two-sided $t$-test. Boxes represent median (middle) and first/third quartile (bottom and top, respectively) values. Box whiskers represent the most extreme values within 150% of interquartile range. Each dot represented one sample. **C** Association between objective response rate (ORR, percentage) and signature scores at cutoff mean values, with $P$ values calculated by two-sided Fisher's exact test. The shaded bar plots represent high/low signatures (Angio, green; Immune, purple; Myeloid, red) ($n$ = 13). **D** Forest plots of response odds ratios (ORs) and confidence interval (CIs) for signature[High] versus signature[Low] populations within Angio, Immune, and Myeloid signatures. All OR and CI values for response were extracted from logistic regression models. **E** Representative image of CD31 IHC staining in patients. Scale bar is 1 mm or 100 μm. **F** Quantification of CD31 density by immunohistochemistry and association with response to camrelizumab combined with apatinib in patients with first-line platinum-resistant (cohort 1, $n$ = 11 NRs and 22 Rs) or PD-1 inhibitor resistant (cohort 2, $n$ = 14 NRs and $n$ = 16 Rs) recurrent/metastatic nasopharyngeal carcinoma. $P$ values were calculated by two-tailed, Mann–Whitney $U$ test. **G** Supervised clustering by response of MCP-counter scores for $z$-score normalization in baseline tumor specimens ($n$ = 9 Rs and 7 NRs). NK cells, natural killer cells. $P$ values were calculated by two-sided Mann–Whitney $U$ test. **H** Quantification of CD20, CD4 and CD8 by immunohistochemistry and association with response in cohort 2 ($n$ = 17 NRs and $n$ = 13Rs). $P$ values were calculated by two-tailed, Mann–Whitney $U$ test. Responders defined as having complete or partial response by RECIST 1.1 and non-responders as having less than a partial response. All data in box and whiskers plots (4B, 4F, and 4H) are represented as median value, the first and third quartile, maximum and minimum; each dot represented one sample. Source data are provided as a Source Data file.

who did not (median: 20.6 vs. 29.0 months). This result corresponds with previous literature[46], indicating that the interval between completing radiotherapy and initiating anti-angiogenic drugs may increase the likelihood of necrosis. Therefore, caution is warranted when using a combination of camrelizumab and apatinib in the future. Only two patients occurred rash and one patient suffered grade 3 myocarditis. The occurrence of hand-foot syndrome, hypertension, proteinuria, and neutropenia (no febrile neutropenia occurred) were likely associated with apatinib[24]. We observed that the incidence of hand-foot syndrome at grade 3 or greater was lower than that of apatinib monotherapy[20,49], which might be attributable to the two-thirds dose reduction of apatinib in this combination in our study. The occurrence of other TRAEs, including fatigue, increased alanine aminotransferase, increased aspartate aminotransferase, hyperbilirubinemia, and laboratory abnormalities, might be associated with the combination treatment.

A key finding from cohort 1 of our study is that the immune microenvironment prior to treatment in baseline tumor tissues appears to drive the clinical activity of camrelizumab plus apatinib in RM-NPC. A consistent trend of increasing efficacy with increasing levels of PD-L1 and KDR expression both in cohort 1 and cohort 2. The positive and negative expression of PD-L1 and KDR and their associations with ORR in both cohorts suggest a strong interplay between the immune reaction and anti-angiogenesis in RM-NPC. Notably, our gene expression analysis found that the B cell-related gene signature further reinforced the clinical significance of B cells, indicating that B cell had a significant interaction with T cells in the tumor microenvironment (TME). More robust responses to the combination of camrelizumab and apatinib occur in patients whose tumors have higher levels of TLS, consistent with other malignancies treated with immune checkpoint inhibitor monotherapy[50–56]. Given that there is a twofold ORR when RM-NPC is treated with camrelizumab and apatinib compared with camrelizumab monotherapy, the mechanism needs further exploration. Our study also provides mechanistic insights into how anti-VEGF may augment antitumor immunity and enhance anti-PD-L1 immunotherapy. Multiple preclinical hypotheses have been speculated for VEGF-mediated immunosuppression, including: (1) increasing T cell exhaustion and decreasing effector function; (2) impairment of T cell priming by suppression of DC maturation; (3) promotion of Treg infiltration and proliferation; and (4) promotion of polarization of tumor-associated macrophages toward a more immunosuppressive phenotype[57–59]. We found that anti-VEGF seemed to synergize with anti-PD-L1 in increasing the number of CD8 + T cells and conventional dendritic cells. Our study provides clinical evidence of how angiogenesis inhibitors may augment antitumor immunity and further enhance the efficacy of anti-PD-L1. However, to our knowledge, by integration of transcriptome, MCP analysis and in situ digital

pathology analysis, we found that patients with higher levels of TLS and B lineage[high] with high response and longer PFS in RM-NPC patients, which was consisted with previous published hypothesis that anti-angiogenic agents can overcome endothelial cell anergy, increase the number of infiltrating T cell into tumor, and induced the formation of tertiary lymphoid structures and high endothelial venules (HEVs)[27,60,61]. However, in the current study, how anti-VEGF induces tertiary lymphoid and the distinct mechanisms through which each subset of B cells contributes need further research. Compared with the patient in cohort 1, MCP analysis found that neutrophils in TME were associated with PD-1 inhibitor resistance, which was consistent with recently published literature[62,63]. We continued to find that a high angiogenesis signature score, but not an immune or myeloid signature, was associated with a high response in cohort 2. The vasculature of tumors is highly abnormal and dysfunctional. Consequently, immune effector cells have an impaired ability to penetrate solid tumors and often exhibit compromised functions. Patients in cohort 2 received apatinib monotherapy in the first two weeks, which is conducive to the normalization of blood vessels. Normalization of the tumor vasculature can enhance tissue perfusion and improve immune effector cell infiltration, leading to immunotherapy potentiation.

Lastly, intrinsic resistance and delayed resistance to anti-PD-1 and anti-VEGF combination therapy showed different evolutionary patterns during relapse. The increase of fibroblasts was the main reason for relapse of patients with intrinsic resistance. However, for the delayed resistance patients, the B cell density in the tissues decreased, while the number of cytotoxic T cells and fibroblasts increased, indicating that the decrease of B cells is the decisive inducement for the relapse. Without B cells' synergetic promotion, the increased cytotoxic T cells cannot perform the function of killing tumor cells. Exhaustion may also be the reason why cytotoxic T cells do not perform their functions; this requires more detailed evidence from subsequent studies. An increase of fibroblasts was also observed in delayed resistance when the tumor relapsed, which suggests that the induction of fibroblasts is common consequence of camrelizumab plus apatinib therapy.

This study has several limitations. First, it had a small sample size given the rarity of the disease. Second, the trial design of this exploratory trial did not allow us to assess whether antitumor activity of apatinib occurred primarily through a direct effect or by reversal of primary PD1 antagonists' resistance. We did not compare the efficacy of this apatinib-camrelizumab regimen with those of anti-PD-1/PD-L1 monotherapies or more recent combined chemoimmunotherapy in this study. Further phase III randomized controlled trials are needed to directly compare apatinib in combination with camrelizumab versus camrelizumab monotherapy or with chemotherapy in treating RM NPC patients as the second or later line treatment therapy. In addition, the

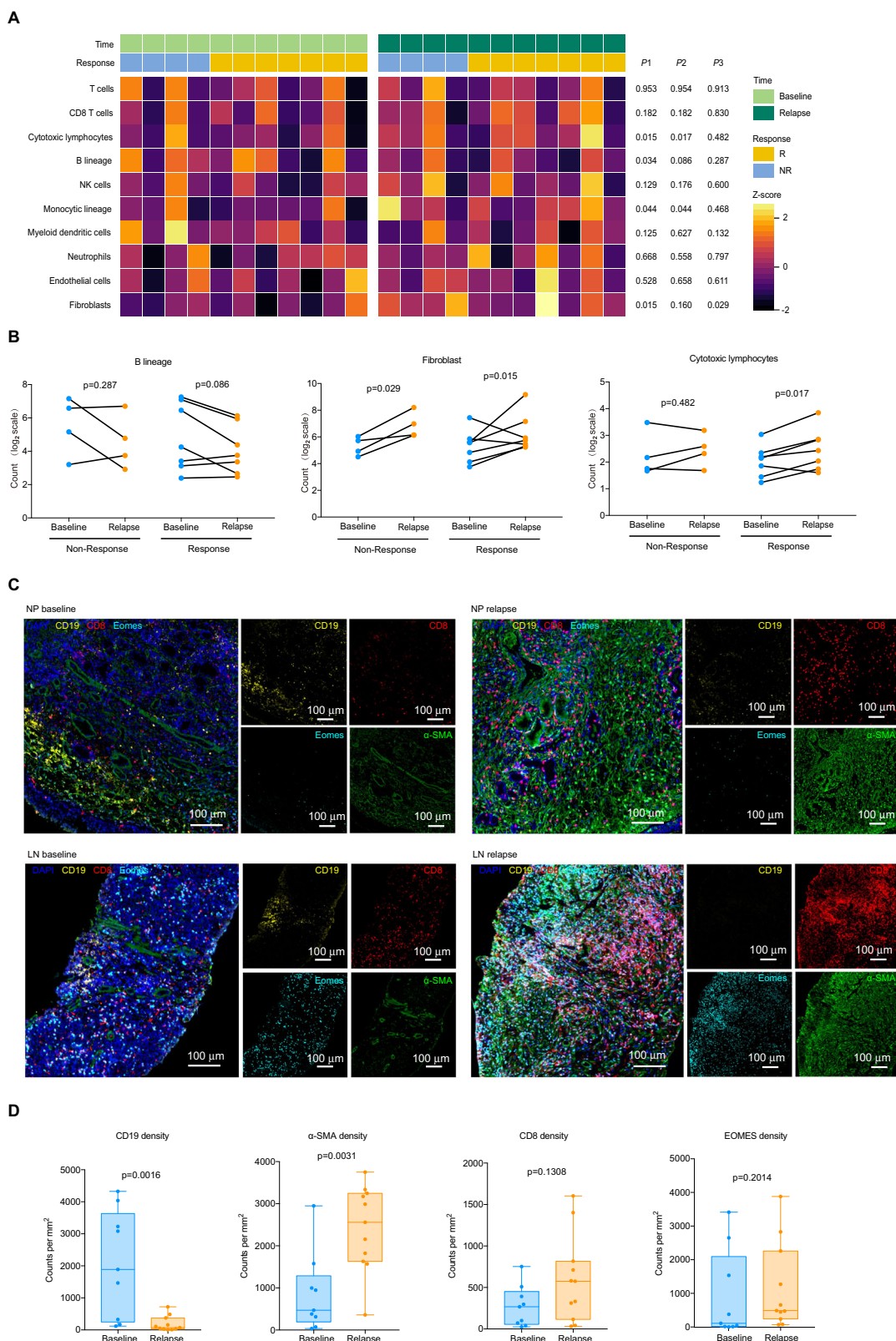

reliability and interpretability of biomarker selection may be limited due to the small sample size of sequencing. Although we found that the tertiary lymph structure was related to therapeutic efficiency, we did not detect the detailed characteristics of the tertiary lymph structure, such as its maturity and heterogeneity, which need to be confirmed in future research. Finally, for this study, the number of front-line treatments included in the population was not completely consistent, and

the treatment plans for front-line therapy varied. Aside from discrepancies in chemotherapy plans, some patients also received anti-EGFR drugs, resulting in significant heterogeneity among the populations studied. Future studies will require further refinement of the characteristics of the study population to conduct large-scale studies.

In conclusion, camrelizumab plus apatinib showed promising anti-tumor activity with a manageable safety profile in platinum-

**Fig. 5 | B cells and fibroblasts have a significant impact on clinical response of camrelizumab plus apatinib treatment. A**. Dynamic changes in cell populations of baseline and disease progression after treatment with camrelizumab plus apatinib in 11 patient-matched tissue biopsies in both cohorts (*n* = 7 Rs and 4 NRs) using MCP-counter method. Compared with the tissue at baseline, the *P1*, *P2* and *P3* represented *P* values that calculated in all patients(*n* = 11), responders(*n* = 7) and non-responders(*n* = 4), respectively. *P* values were calculated by two-sided paired *t*-test. **B**. MCP-counter of B lineage fibroblast and cytotoxic lymphocytes in paired (*n* = 11) biopsy specimens (*n* = 7 Rs and 4 NRs). Paired Samples Wilcoxon Signed

Rank Test was applied in representative gene comparison between baseline and relapse. **C**. Multiplex immunofluorescence assay for the following markers: CD19, α-SMA, CD8, EOMES and DAPI. Original magnification, ×20. Scale bar is 100 μm. **D** Quantification of CD19, α-SMA, CD8 and EOMES density by multiplex immunofluorescence assay (*n* = 9 baseline and 11 relapse). *P* values were calculated by two-tailed, Mann–Whitney *U* test. All data in box and whiskers plots (5D) are represented as median value, the first and third quartile, maximum and minimum. Source data are provided as a Source Data file.

resistant and in PD1 antagonist-resistant RM-NPC. Patients with high expression of PD-L1, KDR and B-cell-related gene signatures will be important to predict which subsets of patients are more likely to benefit from this combination treatment strategy. This combination could be a potential second-line treatment option for patients with RM NPC and warrants phase 3 trials to validate the potential benefits of this regimen.

## Methods

### Study design and participants

This singlearm, openlabel, phase 2 trial was done at a single cancer center in Guangzhou, China. The same inclusion criteria would be applied to both cohorts, but the patient selection criteria differed: cohort 1 included patients with platinum-resistant RM-NPC without received immune checkpoint inhibitors for recurrent or metastatic disease; cohort 2 included patients with immune checkpoint inhibitors-resistant RM-NPC. The first and last patients were officially enrolled in cohort 1 on 8 September 2020, and 30 August 2021, respectively. In cohort 2, patients were enrolled beginning on 2 November 2020, and ending on 7 September 2021.

Eligible patients were aged 18–75 years with an Eastern Cooperative Oncology Group (ECOG) performance status of 0–1, histologically or cytologically confirmed with RM-NPC, not suitable for local treatment, who progressed after receiving first-line platinum-based chemotherapy (cohort 1) or immune checkpoint inhibitor immunotherapy with or without platinum-based chemotherapy (cohort 2), and who presented with measurable tumor lesions assessed by the Response Evaluation Criteria in Solid Tumors (RECIST; version 1.1). Further inclusion criteria were adequate organ function as determined by: Absolute neutrophil count (ANC) ≥ 1.5 × 109/L; Platelet count ≥75 × 109/L; Hemoglobin ≥9 g/dL; serum total bilirubin (TBIL) ≤ 1.5 times the upper limit of normal (ULN); alanine aminotransferase (ALT) and aspartate aminotransferase (AST) ≤ 2.5 × upper limit of normal (ULN) (for subjects with liver metastases, TBIL ≤ 3 × ULN; ALT and AST ≤ 5 × ULN); Creatinine ≤1.5 × ULN or creatinine clearance rate ≥ 50 ml/min (Cockcroft-Gault formula); serum albumin ≥28 g/L; Thyroid-stimulating hormone (TSH) levels ≤1 × ULN; INR, APTT ≤ 1.5 × ULN, and a life expectancy of at least 3 month. Eligible patients were required to provide tumor tissue samples for biomarker analysis. Patients with any active autoimmune disease or history of autoimmune disease, a history of severe bleeding or any bleeding events with a serious grade of 3 or more, MRI showed that the tumor may have invaded important blood vessels or nasopharyngeal necrosis, or those who were previously treated with VEGFR inhibitors (antiangiogenic smallmolecule tyrosine kinase inhibitors, or antiangiogenic monoclonal antibodies) were excluded. Full eligibility and exclusion criteria are included in the study protocol (available in the Supplementary Information file). The trial was approved by the Research Ethics Board of Sun Yat-sen University Cancer Center and was done in accordance with the Declaration of Helsinki. All patients provided written informed consent.

### Procedures

Patients in cohort 1 received intravenous camrelizumab 200 mg every 3 weeks plus oral apatinib 250 mg daily, and patients in cohort 2

received apatinib monotherapy in the first two weeks to modify the immune-resistant microenvironment, and then they were administered camrelizumab plus apatinib. The doses were chosen on the basis of previous published phase I studies in advanced cancers[14,20]. Adverse events were continuously monitored. To continuously detect toxicity[64], a Pocock-type threshold was employed with a toxicity probability of 50% and an early stopping probability of 0.05. Treatment was continued until disease progression, unacceptable toxicity, or withdrawal of consent. Three weeks were considered a treatment cycle. Patients received six cycles of consolidation therapy if a complete or partial response was reported. Dose reduction of camrelizumab and apatinib was not permitted, but dose interruption was permitted if adverse events occurred that were not relieved by supportive care[14]. If patients suffered with grade 3 or worse hematological treatment-related adverse events or with nonhaematological treatment-related adverse events up to grade 2 or worse, drug administration of camrelizumab or apatinib were interrupted until the hematological treatment-related adverse events recovered to grade 2 or better, or non-hematological treatment-related adverse events recovered to grade 1 or better. The first dose reduction of apatinib was to 250 mg once per day with 2 days on and 1 day off, and an additional reduction level was to 250 mg once per day every other day if the adverse event was not relieved[18,19]. Camrelizumab or apatinib were permanently discontinued if the dose interruption exceeded 8 weeks or 4 weeks, respectively. Detailed dose interruptions and discontinuations are presented in the protocol. The severity of necrosis was categorized into different grades, with grade 2 indicating local wound care and medical intervention (e.g., dressings or topical medications). Grade 3 indicated a need for operative debridement or other invasive interventions such as tissue reconstruction, flap or grafting. Grade 4 indicated life-threatening consequences that require urgent intervention, and Grade 5 signified death[46].

Baseline assessment was done within 14 days before treatment, which included thoracic contrast-enhanced CT and abdominal-pelvic and head neck contrast-enhanced CT or MRI scans; electrocardiogram or echocardiography; biochemical, hematological, virological, endocrinological, and urine and feces examinations; and archival tumor tissue assessments. Tumor response was assessed by investigators, according to RECIST version 1.1, used CT and MRI scans every 6 weeks for the first 12 cycle, and every 12 weeks thereafter. Complete or partial response was confirmed by subsequent scans at least 28 days apart. Routine blood and hepatic, renal function examinations, hematological, biochemical, endocrinological, and urine and fecal examinations were done every cycle. Adverse events were monitored before each drug administration and at each examination throughout the treatment process, and at 30 days (30, 60, and 90 days for serious adverse events) after the last dose of the study drug. Adverse events were graded according to the National Cancer Institute Common Terminology Criteria for Adverse Events version 5.0.

### Outcomes

The primary endpoint was the proportion of patients achieving an objective response according to RECIST version 1.1, which included patients with measurable disease who had a complete or partial

response. Secondary endpoints were progression-free survival, duration of response, proportion of disease control, and safety. Progression-free survival was defined as the interval from the start of treatment to disease progression or death for any cause (whichever occurred first) or the last progression-free survival assessment for patients alive without progression. Duration of response was assessed in patients who achieved a response and was defined as the time from the date of the first documented response until the date of documented progression or death from any cause. Disease control was defined as the proportion of patients who achieved complete response, partial response or stable disease. The overall survival was defined as the time from treatment initiation to death for any reason.

## Statistical analysis

We used Simon's two-stage design. For cohort 1, the previously reported data indicated that the objective response of PD-1 monotherapy in platinum-resistant NPC was about 25%[12–14]. We initially expected that the objective response for apatinib combined with camrelizumab would be 50%. This had a one-sided type I error rate of 5% and a power of 80%. In the first stage, 9 patients were accrued. If more than two responders were observed, an additional 15 patients would be accrued to the second stage. The study was considered positive if more than nine responders were observed among the 24 patients. Considering 10% patients loss to follow-up, at least 27 patients were enrolled in cohort 1.

For cohort 2, although the previously reported data indicated that the response rate of apatinib monotherapy in platinum-resistant NPC was about 30%[30,31], the proportion of patients achieving an objective response for PD-1 blockade-resistant RM-NPC with apatinib monotherapy was not well defined. Therefore, in cohort 2, we assumed that the proportion of patients achieving an objective response of 20% with apatinib monotherapy for PD-1 blockade-resistant RM-NPC, which was less than the objective response of 30% for the PD-1 blockade-naive RM-NPC. Assuming that camrelizumab plus apatinib lead to ORR achieved 45% for PD-1 inhibitor resistant RM-NPC, with a one-sided α error of 5% and a power of 80%, at least 22 patients should be enrolled. Ten patients were counted in the first phase. 12 more patients will be included in the second phase if more than two responses are seen in the first stage. If out of 22 patients, more than seven responded, the trial was deemed successful. After considering a 10% dropout rate of patients, at least 25 patients were enrolled in cohort 2. The full analysis set included all the patients who received at least one dose of camrelizumab and apatinib. The efficacy analysis set included the patients who received at least one post-baseline efficacy assessment. Objective response rate, progression-free survival, and overall survival as well as disease control rate, were analyzed both in the full analysis set and the efficacy analysis set. Duration of response, tumor shrinkage rate, and time to response were analyzed in the efficacy analysis set. Patients who received the camrelizumab and apatinib and had at least one post-baseline safety assessment been included in the safety set. Duration of response, and time to response, median progression-free survival and overall survival, were estimated using the Kaplan–Meier method, and their 95% CIs were calculated using the Brookmeyer-Crowley method. Progression-free survival rate at 6 months and 12 months, overall survival rate at 6 months and 12 months, and post-hoc analysis of 12-month duration of response rate were estimated by the Kaplan-Meier method, and the corresponding 95% CIs were calculated by log–log transformation method. We also did post-hoc subgroup analyses according to gender (male vs female), age (<60 years vs ≥60 years), WHO classification (non-keratinizing differentiated vs non-keratinizing undifferentiated), ECOG score s (0 vs 1), liver metastases (yes vs no), smoking history (yes vs no), previous lines of therapy for advanced disease s (one vs >two or more), baseline plasma EBV DNA status (<10,000 copies/ml vs ≥ 10,000 copies/ml), previous radiotherapy(yes

vs no), previous EGFR inhibitor treatment ((yes vs no), PD-L1 expression and VEGF receptor 2 (KDR) expression at baseline.

The statistical comparison between responder and non-responder groups for a given continuous variable was performed using two-sided Mann–Whitney U-test. Univariable and multivariable analysis predicting response to ICB was performed using logistic regression modeling. Biological replicates are indicated in the individual figure legends. Technical replicates were constrained to $n = 1$ per time point, owing to limited tissue availability in patient-derived samples as well as prioritization for multiple studies. The trial was not randomized, and investigators were not blinded to allocation during experiments or outcome assessment unless stated otherwise. Hazard ratios for progression-free survival and overall survival were estimated using Cox proportional hazard models. The data in this study were collected using the EpiData 3.1. We used SPSS (version 22.0) and R (version 4.2.1) for analyses. Considering that the purpose of this trial was to evaluate the efficacy and safety of carrilizumab and apatinib in first-line platinum-resistant or PD-1 inhibitor-resistant RM-NPC, and due to the target population, the reference ORRs used to sample size calculation, inclusion criteria and drug administration schedule was different between cohort 1 and cohort 2, we have registered the two cohorts of the trial as two separate trials on clinicaltrials.gov in order to avoid misunderstanding and make it easier to understand for the readers. The trials were registered on Sep.14, 2020, with Clinical-Trials.gov, with identifiers NCT04547088 for cohort 1 and NCT04548271 for cohort 2.

## Tumor sample collection and preparation

Fresh tumor biopsies were retrieved from NPC patients and divided into two parts on the premise of informed consent. One part of tumor biopsies was immediately put into liquid nitrogen for rapid freezing for subsequent RNA sequencing; the other part was embedded in paraffin and sectioned for immunohistochemical staining and multiple immunofluorescence staining.

## RNA extraction and RNA sequence

Total RNA was extracted from snap-frozen tumor specimens. RNA purity was checked using the NanoPhotometer ® spectrophotometer (IMPLEN, CA, USA). RNA integrity was assessed using the RNA Nano 6000 Assay Kit of the Bioanalyzer 2100 system (Agilent Technologies, CA, USA). 1 μg RNA per sample was used as input material for the RNA sample preparations. Sequencing libraries were generated using NEBNext® UltraTM RNA Library Prep Kit for Illumina® (NEB, USA) following manufacturer's recommendations and index codes were added to attribute sequences to each sample. Purified mRNA was randomly fragmented using divalent cations under elevated temperature in NEBNext First Strand Synthesis Reaction Buffer(5X). Using mRNA as template, double-stranded cDNA was synthesized by reverse transcription. cDNA fragments of around 200 bp were selected (AMPure XP beads) and purified (Agilent Bioanalyzer 2100 system) to ensure the quality of the library. The clustering of the index-coded samples was performed on a cBot Cluster Generation System using TruSeq PE Cluster Kit v3-cBot-HS (Illumia) according to the manufacturer's instructions. After cluster generation, the library preparations were sequenced on an Illumina Novaseq platform and 150 bp paired-end reads were generated.

## RNA-seq data processing and quality check

Raw data (raw reads) of fastq format were firstly processed through in-house perl scripts. At the same time, Q20, Q30, and GC content the clean data were calculated. All the downstream analyses were based on the clean data with high quality. Reference genome and gene model annotation files were downloaded from genome website. Hisat2 v2.0.5 was used to build the index of the reference genome and to align paired-end clean reads to the reference genome.

## Gene expression quantification and normalization

featureCounts v1.5.0-p3 was used to count the reads numbers mapped to each gene. And then FPKM of each gene was calculated based on the length of the gene and reads count mapped to this gene. FPKM, expected number of Fragments Per Kilobase of transcript sequence per Millions base pairs sequenced, considers the effect of sequencing depth and gene length for the reads count at the same time, and is currently the most commonly used method for estimating gene expression levels.

## Identification of DEGs

Differential expression analysis of two conditions/groups (two biological replicates per condition) was performed using the DESeq2 R package (1.16.1). DESeq2 provide statistical routines for determining differential expression in digital gene expression data using a model based on the negative binomial distribution. The resulting *P*-values were adjusted using the Benjamini and Hochberg's approach for controlling the false discovery rate. Genes with an adjusted *P*-value < 0.05 found by DESeq2 were assigned as differentially expressed.

## Deconvolution of the cellular composition with MCP-counter

The R package software MCP-counter was applied to the normalized $\log_2$-transformed FPKM expression matrix to produce the absolute abundance scores for eight major immune cell types (CD19 + B cells, CD8 + T cells, CD4 + T cells), epithelial cells, and fibroblasts. The deconvolution profiles were then hierarchically clustered and compared across response and treatment groups.

## Pathway enrichment analyses

The network-based pathway enrichment analysis was performed using DEGs across responder and non-responder groups in the bulk-tissue RNA-seq data. In the bulk-tissue, the differentially expressed genes that had a *q* < 0.05 and $\log_2$-trans- formed fold change >1.5 or <−1.5 were selected as input for network-based pathway enrichment analysis.

## Immunohistochemistry and multiplex immunohistochemical staining

A retrospective study was performed on formalin-fixed, paraffin-embedded (FFPE) tumor tissues of NPC and metastatic liver, lung and lymph gland. Multiplex immunohistochemical staining was performed as follow steps, 4 μm sections from full FFPE blocks of tumor tissues were sectioned, dewaxed, and fixed with 10% neutralized formaldehyde. Then, antigen was retrieved using heated Tris- EDTA buffer (pH 8.0 or pH 9.0) for 2.5 min. Each section was subjected to four successive rounds of antibody staining after the initial establishment of staining conditions for each individual primary antibody and successive optimization. Each staining step consisted of blocking with 20% normal goat serum/fetal bovine serum in PBS and incubation with primary antibodies, followed by biotinylated anti-mouse/anti-rabbit secondary antibodies and streptavidin–horseradish peroxidase (HRP) substrate. Then, the immunoreactive stains were visualized using tyramide signal amplification (TSA) with fluorophores Opal 480, 520, 570, and 690 diluted in 1×Plus Amplification Diluent. Finally, the Ab–TSA complexes were stripped in heated Tris-EDTA buffer (pH 8.0 or pH 9.0) for 2.5 min. Nuclei were counterstained with 4′, 6- diamidino-2- phenylindole, dihydrochloride (DAPI) and sections were mounted using Perma Fluorfluorescence mounting medium (PANOVUE). The same procedure without primary antibodies was used as a negative control. The multiplex immunohistochemical staining results were scored based on the percentage of the number of cell subsets. Computer recognition software used indicated molecules to identify subsets of cells automatically and count them. Cell quantification was performed across whole tumor sections using Halo analysis software (PANOVU). The antibodies used for IHC staining are CD4 (ZM-0418, ZSbio, Clone: EP204); CD19 (ZM-0038, ZSbio, Clone: UMAB103); CD8 (ZA-0508, ZSbio, Clone: SP16); CD3 (ZM-0417 ZSbio Clone: LN10); Eomes (ab183991, Abcam, 1:200); α-SMA (ab7817, Abcam, 1μg/mL); KDR (ab2349, Abcam, 1:100); PD-L1 (ab205921, Abcam, 2μg/mL); CD31(ab28364, Abcam, 1:50); IL8 (94407 T, Cell Signaling Technology, 1:100); VEGFA (ab52917, Abacm, 1:100); c-KIT (ab32363, Abcam, 1:400); SRC (ab109381,Abacam, 1:400).

## TLS quantification

TLSs were qualified and quantified using both H&E and CD19 + CD3+ IHC staining. Structures were identified as aggregates of lymphocytes having histological features with analogous structures to that appearing in the tumor area. For the current study, criteria used for the quantification of TLS is mean area.

## Reporting summary

Further information on research design is available in the Nature Portfolio Reporting Summary linked to this article.

## Data availability

The trial protocol is available as Supplementary Note 1 in the Supplementary File.

The raw sequencing data can be accessed through GSA under the accession code HRA004738. Sequencing data are available under restricted access. Access can be obtained by completing the application form via GSA-Human System[65,66] and/or by contacting the corresponding authors. Clinical data are not publicly available due to involving patient privacy, but can be accessed on request from the corresponding author Hai-Qiang Mai for 10 years; individual de-identified participant data will be shared. All requests for data will be reviewed by the leading clinical site Sun Yat-Sen University Cancer Center and the study sponsor, Jiangsu Hengrui Pharmaceuticals, to verify if the request is subject to any intellectual property or confidentiality obligations. Requests for access to the patient-level data from this study can be submitted via email to *maihq@sysucc.org.cn* with detailed proposal for approval. A signed data access agreement with the sponsor is required before accessing the shared data. The full IHC/MIHC dataset could also be shared upon request to the corresponding author, Hai-Qiang Mai, via email at *maihq@sysucc.org.cn*.

Source data are provided with this paper. The remaining data are available within the Article, Supplementary Information, and Source Data. Source data are provided with this paper.

## Code availability

No novel code/algorithm were used in this study. All code used in this study for different expression genes, GO and KEGG enrichment, MCP–counter, and gene set variation analysis (GSVA) is available from the corresponding author upon request.

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

## Acknowledgements

This study was funded by grants from the National Key Research and Development Program of China (2022YFC2505800, L.Q.T.; 2022YFC2705005, H.Q.M.), National Natural Science Foundation of China (82173287, H.Q.M. and 82073003, L.Q.T.), Guangdong Basic and Applied Basic Research Foundation (2021B1515230002, H.Q.M.). The funders had no role in study design, data collection and analysis, or manuscript writing.

## Author contributions

Conception and design: H.Q.M., L.Q.T., L.Y., and Q.Y.C. designed the study. Provision of study materials or patients: L.Y., G.D.J., X.F.L., S.Y.X., S.S.G., D.F. L., L.T.L., D.H.L., Y.F.L, S.W.D., L.G., M.S.Z., X.Y.C., S.L.L., X.S.S., X.Y.L., S.C.L., Q.Y.C., L.Q.T., and H.Q.M. Data analysis and interpretation: L.Y., G.D.J., X.F.L., S.Y.X., S.S.G., D.F. L., Q.Y.C., L.Q.T., and H.Q.M. Manuscript writing: all authors.

## Competing interests

The authors declare no competing interests.

## Additional information

Li Yuan[1,2,5], Guo-Dong Jia[1,2,5], Xiao-Fei Lv[1,3,5], Si-Yi Xie[1,2,5], Shan-Shan Guo[1,2,5], Da-Feng Lin[1,2,5], Li-Ting Liu[1,2], Dong-Hua Luo[1,2], Yi-Fu Li[1,2], Shen-Wen Deng[1,2], Ling Guo[1,2], Mu-Sheng Zeng ®[1], Xiu-Yu Cai[1,4], Sai-Lan Liu[1,2], Xue-Song Sun[1,2], Xiao-Yun Li[1,2], Su-Chen Li ®[1,2], Qiu-Yan Chen[1,2,6], Lin-Quan Tang[1,2,6] & Hai-Qiang Mai ®[1,2,6] ✉

[1]Sun Yat-sen University Cancer Center, State Key Laboratory of Oncology in South China, Collaborative Innovation Center for Cancer Medicine, Guangdong Key Laboratory of Nasopharyngeal Carcinoma Diagnosis and Therapy, 651 Dongfeng Road East, Guangzhou 510060, People's Republic of China. [2]Department of Nasopharyngeal Carcinoma, Sun Yat-sen University Cancer Center, 651 Dongfeng Road East, Guangzhou 510060, People's Republic of China. [3]Department of Medical Imaging, Sun Yat-sen University Cancer Center, 651 Dongfeng Road East, Guangzhou 510060, People's Republic of China. [4]Department of General Internal Medicine, Sun Yat-sen University Cancer Centre, 651 Dongfeng Road East, Guangzhou 510060, People's Republic of China. [5]These authors contributed equally: Li Yuan, Guo-Dong Jia, Xiao-Fei Lv, Si-Yi Xie, Shan-Shan Guo, Da-Feng Lin. [6]These authors jointly supervised this work: Qiu-Yan Chen, Lin-Quan Tang, Hai-Qiang Mai. ✉e-mail: maihq@sysucc.org.cn

