## [Peer Review File · Nature Communications]

Camrelizumab Combined With Apatinib in Patients With First-line Platinum-resistant or PD-1 inhibitor Resistant Recurrent/Metastatic Nasopharyngeal Carcinoma: a Single-arm, Phase 2 trialEditorial Note: This manuscript has been previously reviewed at another journal that is not operating a transparent peer review scheme. This document only contains reviewer comments and rebuttal letters for versions considered at *Nature Communications*.

REVIEWERS' COMMENTS

Reviewer #1 (Remarks to the Author):

[none]

Reviewer #2 (Remarks to the Author):

the authors have provided details replies to most of the reviewers' questions from the previous round of review.

However since the initial submission and review of the current article, Ding et al has published in JCO 2023 epub 3rd Feb , a study using the identical regimen in a nearly identical cohort of recurrent metastatic NPC patients from the same institution as this study which reported similar clinical response rates which were the primary endpoints of both studies. The main secondary endpoints of Toxicities were also comparable. The trial period of the published study preceded this one and both studies were registered with ClinicalTrials.gov

Hence the authors claim in the manuscript that "this is the first trial to evaluate....."(lines 437-9) would not be applicable, and the publication of the earlier study will need to be addressed and the results of the current study put in the appropriate perspective.

Reviewer #3 (Remarks to the Author):

[none]

Reviewer #4 (Remarks to the Author):

Authors have addressed my comments/concerns.

Reviewers' comments:

Reviewer #2 (Remarks to the Author):

the authors have provided details replies to most of the reviewers' questions from the previous round of review.

However since the initial submission and review of the current article, Ding et al has published in JCO 2023 epub 3rd Feb , a study using the identical regimen in a nearly identical cohort of recurrent metastatic NPC patients from the same institution as this study which reported similar clinical response rates which were the primary endpoints of both studies. The main secondary endpoints of Toxicities were also comparable. The trial period of the published study preceded this one and both studies were registered with ClinicalTrials.gov

Hence the authors claim in the manuscript that "this is the first trial to evaluate....."(lines 437-9) would not be applicable, and the publication of the earlier study will need to be addressed and the results of the current study put in the appropriate perspective.

Reply: Thank you for the reviewer's kind comment. The published research results of Ding et al [1] in JCO (ORR: 65.5%, 95% CI, 51.9 to 77.5) were similar to the ORR results of our cohort 1 (65.0%, 95% CI, 49.6 to 80.4). Furthermore, Ding et al's study reported on grade 3 or higher treatment-related adverse events that were similar to those we reported (58.6% vs. 65.3%), confirming the effectiveness and feasibility of combining anti-angiogenic targeted therapy with immunotherapy in platinum-resistant, later-line recurrent nasopharyngeal carcinoma. Additionally, our cohort 2 reported promising results for the combination of camrelizumab and apatinib after PD-1 resistance (ORR: 34.3%), providing new evidence for patients with recurrent nasopharyngeal carcinoma resistant to PD-1 inhibitors. We have revised the earlier study's discussion on page 15, line 444.

Reference: [1] Ding X, Zhang WJ, You R, et al. Camrelizumab Plus Apatinib in Patients With Recurrent or Metastatic Nasopharyngeal Carcinoma: An Open-Label, Single-Arm, Phase II Study. J Clin Oncol. 2023;41(14):2571-2582.